# Eyeless uncouples mushroom body neuroblast proliferation from dietary amino acids in *Drosophila*

Conor W Sipe, Sarah E Siegrist*

Department of Biology, University of Virginia, Charlottesville, United States

**Abstract** Cell proliferation is coupled with nutrient availability. If nutrients become limited, proliferation ceases, because growth factor and/or PI3-kinase activity levels become attenuated. Here, we report an exception to this generality within a subpopulation of *Drosophila* neural stem cells (neuroblasts). We find that most neuroblasts enter and exit cell cycle in a nutrient-dependent manner that is reversible and regulated by PI3-kinase. However, a small subset, the mushroom body neuroblasts, which generate neurons important for memory and learning, divide independent of dietary nutrient conditions and PI3-kinase activity. This nutrient-independent proliferation is regulated by Eyeless, a Pax-6 orthologue, expressed in mushroom body neuroblasts. When Eyeless is knocked down, mushroom body neuroblasts exit cell cycle when nutrients are withdrawn. Conversely, when Eyeless is ectopically expressed, some non-mushroom body neuroblasts divide independent of dietary nutrient conditions. Therefore, Eyeless uncouples MB neuroblast proliferation from nutrient availability, allowing preferential neurogenesis in brain subregions during nutrient poor conditions.

DOI: https://doi.org/10.7554/eLife.26343.001

## Introduction

Quiescence versus proliferation decisions require coordination of stem cell-intrinsic factors with extrinsic factors, local and systemic, that vary in response to changing animal physiology (*Cheung and Rando, 2013*; *He et al., 2009*; *Orford and Scadden, 2008*). Nutrient availability is an important extrinsic factor as nutrients provide the building blocks for macromolecular biosynthesis that drives cell growth and proliferation (*Chantranupong et al., 2015*; *Lunt and Vander Heiden, 2011*). Here, we use *Drosophila* neuroblasts (NBs)(*Brand and Livesey, 2011*; *Doe, 2008*; *Homem and Knoblich, 2012*) to determine how neural stem cell proliferation decisions are made in response to dietary amino acid availability. NBs enter quiescence at the end of embryogenesis and reactivate soon after freshly hatched larva consume their first complete meal. Developmental quiescence is 'pre-programmed' and all NBs (~100) enter quiescence except for a small subset, which includes the four mushroom body NBs (MB NBs) and one lateral NB in each brain hemisphere (*Ito and Hotta, 1992*; *Truman and Bate, 1988*; *Tsuji et al., 2008*; *Britton and Edgar, 1998*). NB reactivation from quiescence is regulated by a nutritional checkpoint that requires dietary amino acids and is coupled to larval growth by the fat body (*Britton and Edgar, 1998*; *Chell and Brand, 2010*; *Colombani et al., 2003*; *Géminard et al., 2009*; *Sousa-Nunes et al., 2011*). In response to animal feeding, the fat body releases a systemic signal, which induces synthesis and secretion of Dilp-6 (insulin-like peptide 6) from brain glia (*Chell and Brand, 2010*; *Sousa-Nunes et al., 2011*). Dilp-6 in turn activates the insulin-like tyrosine kinase receptor (InR) in nearby NBs (*Chell and Brand, 2010*; *Sousa-Nunes et al., 2011*). InR activates PI3-kinase, a highly conserved regulator of cell growth, which stimulates cellular nutrient uptake via regulation of transmembrane transporters,

*For correspondence:
ses4gr@virginia.edu

**Competing interests:** The authors declare that no competing interests exist.

regulates key metabolic enzymes necessary for macromolecular biosynthesis, and leads to downstream activation of anabolic growth signaling pathways, most notably TOR (*Engelman et al., 2006*).

While amino acids are required to reactivate quiescent NBs, it is unclear whether further dietary amino acid intake is required. MB NBs continue proliferating during the embryonic to larval transition and in the absence of the food-derived systemic signal (*Britton and Edgar, 1998*; *Lin et al., 2013*). Here, we report that NB subtypes have different dietary nutrient requirements for proliferation and that these differences are regulated by cell autonomous, lineage factors.

## Results and discussion

Freshly hatched larvae were fed a complete nutrient diet and then switched to a sucrose-only diet (hereafter referred to as dietary nutrient withdrawal) (*Figure 1A*). EdU, a thymidine analogue, was added to the diet for the final 24 hr to assay NB proliferation in the absence of dietary amino acids. After 24 hr of complete feeding, 84% of central brain NBs, identified based on expression of the transcription factor Deadpan (Dpn) and large cell size, were EdU-positive, indicating NB reactivation from developmental quiescence (*Figure 1A,B,H*) (*Britton and Edgar, 1998*; *Chell and Brand, 2010*; *Sousa-Nunes et al., 2011*). Animals were then switched to a sucrose-only diet, and a reduction in EdU-positive NBs was observed over time, from 77% at 1 day AFW (after food withdrawal) to 3% at 7 days AFW (*Figure 1C–F,H*). This reduction was not due to a change in NB number, suggesting that NBs require dietary amino acids to maintain proliferation (*Figure 1—figure supplement 1A*). Animals at 7 d AFW were reintroduced to a complete diet and after 3 days of refeeding, essentially all NBs were EdU-positive (*Figure 1A,G,H*). We conclude that during early larval stages, most NBs enter and exit cell cycle in a nutrient-dependent manner and, like developmental quiescence, this process is reversible. However, we found that nutrient-arrested NBs in animals at 7 day AFW were larger than quiescent NBs in freshly hatched larvae, and re-enter S phase without subsequent cell size increases (*Figure 1—figure supplement 1B*). This suggests that nutrient-arrest and quiescence are distinct. Nevertheless, both nutrient-arrested and quiescent NBs re-enter cell cycle in response to nutrition, suggesting that common signaling pathways regulate both processes.

Next, to fully reactivate all NBs, freshly hatched larvae were fed a complete diet for 48 hr, and then switched to a sucrose-only diet. Again, a reduction in EdU-positive NBs was observed over time, which was not due to a change in NB number (*Figure 1—figure supplement 1E,F*). At 7 days AFW, 52% of NBs were EdU-positive, while at 14 days AFW, 23% of NBs were EdU-positive (*Figure 1—figure supplement 1F*). Compared to 24 hr fed animals, more NBs remained EdU-positive after dietary nutrient withdrawal, which could be due to increased levels in stored nutrients or to intrinsic differences among NBs between 24 and 48 hr fed animals. Nevertheless, many NBs resumed proliferation after animal refeeding (*Figure 1—figure supplement 1F*).

To better understand regulation of NB proliferation during dietary nutrient withdrawal, we asked whether PI3-kinase is required. PI3-kinase is active during nutrient-rich conditions and is required to reactivate quiescent NBs (*Chell and Brand, 2010*; *Sousa-Nunes et al., 2011*; *Engelman et al., 2006*; *Puig and Tjian, 2006*). During dietary nutrient withdrawal, when PI3-kinase was constitutively activated in NBs (*worGAL4,UASdp110$^{CAAX}$*) (*Albertson and Doe, 2003*; *Brand and Perrimon, 1993*; *Leevers et al., 1996*), NBs were found to incorporate EdU longer, compared to controls (*Figure 2A–E,K,M*). Conversely, when PI3-kinase activity was reduced (*worGAL4,UASdp60*) (*Weinkove et al., 1999*), a reduction in EdU-positive NBs was observed over time compared to controls (*Figure 2F–J,L,M*). Therefore, during early larval stages, NBs exit cell cycle in a PI3-kinase-dependent manner and levels of PI3-kinase activity in response to dietary nutrients are required to maintain NB proliferation.

Unlike other central brain NBs, MB NBs remain large in size and divide continuously during the embryonic to larval transition, suggesting that NB subtypes have different dietary nutrient requirements for proliferation (*Ito and Hotta, 1992*; *Truman and Bate, 1988*; *Britton and Edgar, 1998*; *Lin et al., 2013*). We found that unlike other central brain NBs, the four MB NBs divide continuously after dietary nutrient withdrawal (*Figure 1F,H*, and *Figure 1—figure supplement 1C,D*) (*Britton and Edgar, 1998*; *Lin et al., 2013*). Moreover, we found that MB NB proliferation during dietary nutrient withdrawal is also PI3-kinase independent. PI3-kinase is typically regulated in a nutrient-dependent manner, through Dilp binding to InR, but PI3-kinase can also be regulated

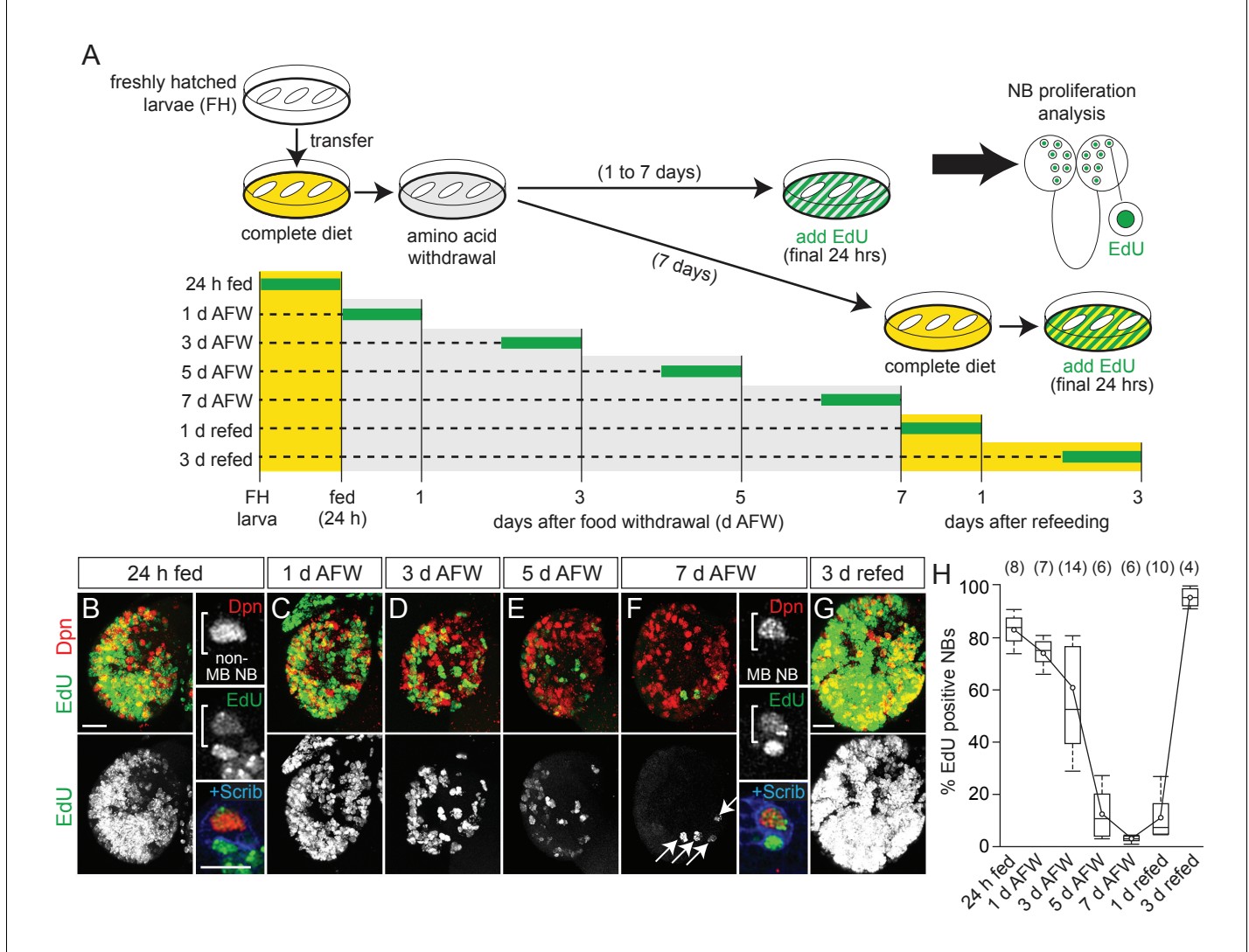

**Figure 1.** NB subtypes respond differently to dietary amino acid withdrawal. (**A**) Experimental protocol to assay NB proliferation during dietary amino acid withdrawal. Freshly hatched (FH) larvae were transferred to a complete diet (yellow). Animals fed for 24 hr to reactivate NBs from quiescence and then transferred to a sucrose-only diet (grey) and maintained for 1–7 days. Twenty-four hours before analysis, EdU (green) was added to the diet to assay NB proliferation. For refeeding experiments, larvae were transferred back to a complete diet (yellow). (**B–G**) Maximum intensity projections of single brain hemispheres, top panel colored overlay with single-channel greyscale image below. Midline is right in this and in all subsequent figures. Scale bar, 20 µm. White arrows designate the four MB NBs. (**B,F**) On the right, single channel greyscale images of NBs at higher magnification with colored overlay below. Scale bar, 10 µm. NBs in white brackets. (**H**) Box plots of the percent EdU-positive NBs per brain hemisphere at the indicated time points. Numbers in parentheses indicate the number of brain hemispheres analyzed at each time point (refer to Materials and methods).

DOI: https://doi.org/10.7554/eLife.26343.002

The following source data and figure supplements are available for figure 1:

**Source data 1.** Quantification of EdU-positive and Dpn-positive cells in control brains during dietary amino acid withdrawal.
DOI: https://doi.org/10.7554/eLife.26343.005

**Figure supplement 1.** Dietary nutrients are required for NB proliferation but do not affect NB number.
DOI: https://doi.org/10.7554/eLife.26343.003

**Figure supplement 1—source data 1.** Quantification of NB number and cell size measurements in control brains fed 24 hr before dietary amino acid removal and refeeding.
DOI: https://doi.org/10.7554/eLife.26343.004

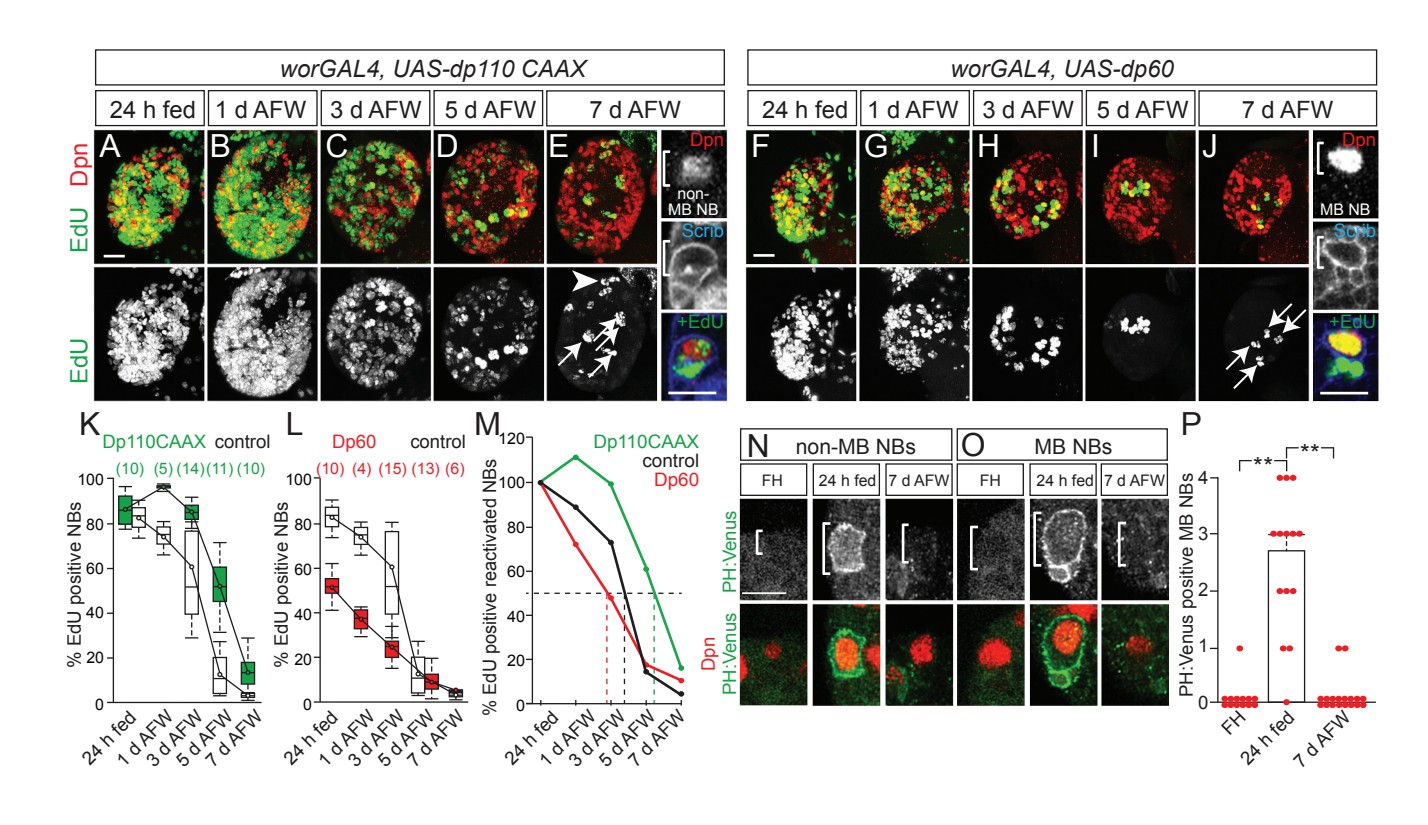

**Figure 2.** MB NBs proliferate in a PI3-kinase-independent manner during dietary amino acid withdrawal, but not non-MB NBs. (A–J) Maximum intensity projections of single brain hemispheres, top panel colored overlay with single-channel greyscale image below. Genotypes listed above and molecular markers to the left. Scale bar, 20 μm. White arrows designate the four MB NBs. Arrowhead in E indicates non-MB NB shown to the right. (E,J) Single channel greyscale images of NBs at higher magnification on the right with colored overlay below. Scale bar, 10 μm. NBs in white brackets. (K,L) Box plots of the percent of EdU-positive NBs per brain hemisphere at the indicated time points. Numbers in parentheses indicate the number of brain hemispheres analyzed at each time point, color corresponds to genotype. (M) Percent of EdU-positive NBs normalized to the percent of reactivated NBs after 24 hr of feeding. Dotted lines indicate the time at which 50% of reactivated NBs are EdU-positive. Single channel greyscale images with color overlay below of non-MB NBs (N) and of MB NBs (O) at indicated time points listed above with reporter and molecular markers listed to the left. White brackets denote NBs. and scale bar (N) equals 10 μm. (P) Histogram of average number of MB NBs positive for membrane PH:Venus fluorescence. p-values are $2.7 \times 10^{-7}$ and $6.9 \times 10^{-9}$, respectively (Student's t-test). Small circles denote primary data.

DOI: https://doi.org/10.7554/eLife.26343.006

The following source data and figure supplements are available for figure 2:

**Source data 1.** Quantification of EdU-positive and Dpn-positive cells in brains expressing *UAS-dp110 CAAX* and *UAS-dp60* in NBs during dietary amino acid withdrawal and quantification of MB NBS in *PH:Venus* animals.
DOI: https://doi.org/10.7554/eLife.26343.009

**Figure supplement 1.** Altered PI3-kinase activity or reduced Alk levels do not affect EdU incorporation in MB NBs or their progeny.
DOI: https://doi.org/10.7554/eLife.26343.007

**Figure supplement 1—source data 1.** Quantification of EdU-positive MB NBs and their EdU-positive progeny in brains of the indicated genotypes.
DOI: https://doi.org/10.7554/eLife.26343.008

independently of dietary nutrient conditions via binding of the low-density lipoprotein-like Jelly Belly to Alk, a tyrosine kinase receptor similar to InR (*Cheng et al., 2011*). At 7 days AFW, when levels of PI3-kinase activity (*worGAL4,UAS-dp60*) or Alk (*worGAL4,UAS-AlkRNAi*) were reduced in NBs, no difference in the number of EdU-positive MB NBs or their EdU-positive progeny was observed compared to controls (*Figure 2J* and *Figure 2—figure supplement 1A–D*). This suggests that MB NBs utilize a PI3-kinase-independent mechanism to maintain proliferation during dietary nutrient withdrawal.

PI3-kinase is a lipid kinase and when active converts PIP2 (phosphatidylinositol-4,5-biphosphate) to PIP3 (phosphatidylinositol-3,4,5-triphosphate). PIP3-rich plasma membrane domains serve as

recruitment sites for pleckstrin homology (PH) domain-containing proteins, including PH:Venus, a PI3-kinase activity reporter (*Doyle et al., 2017*; *Khuong et al., 2013*). After 24 hr of feeding, we observed an increase in the number of non-MB NBs and MB NBs that have PH:Venus along the plasma membrane (data not shown, and *Figure 2N–P*). At 7 days AFW and in freshly hatched larvae, PH:Venus was not detected along MB NB membranes (*Figure 2O,P*). Therefore, dietary nutrient intake is required to initiate and maintain PI3-kinase activity in NBs during early larval stages, even in MB NBs, which can proliferate independent of dietary nutrient conditions and PI3-kinase activity.

MB NBs reside in close proximity to each other and to other central brain NBs (hereafter referred to as non-MB NBs) within a common macro-environment in the central brain, suggesting that quiescence versus proliferation decisions may be regulated cell autonomously. We asked whether Eyeless (Ey), a paired-box homeodomain transcription factor required for MB neuropil formation, is also required for dietary nutrient-independent MB NB proliferation (*Clements et al., 2009*; *Kurusu et al., 2000*; *Noveen et al., 2000*). During complete feeding and following dietary nutrient withdrawal, all MB NBs and some non-MB NBs express Ey at low levels (*Figure 3*) (*Kurusu et al., 2000*; *Noveen et al., 2000*). Ey is also expressed at higher levels in some neurons and glia. However, most non-MB NBs do not express Ey, which led us to ask whether Ey is required for MB NB proliferation during dietary nutrient withdrawal.

We used RNAi to knockdown Ey in MB NBs (*worGAL4,UAS-eyRNAi*) (*Figure 3A,B,E,F,R*). After 24 hr of feeding, all four MB NBs in each brain hemisphere were EdU-positive, as in controls (*Figure 4A,B*). However, at 7 days AFW, the number of EdU-positive MB NBs was reduced in *eyRNAi* animals compared to controls (*Figure 4C,D*). To determine if MB NBs become nutrient-dependent like non-MB NBs when Ey is absent, *eyRNAi* animals at 7 days AFW were reintroduced to a complete diet. After 1 day of refeeding, essentially all MB NBs were EdU-positive (*Figure 4E,F*). Next, we asked whether ectopic Ey expression is sufficient to switch non-MB NB proliferation from nutrient-dependent to nutrient-independent. We expressed a wild-type version of Ey in NBs (*worGAL4, UAS-ey^{WT}*) (*Figures 3G–T* and *4G–N*) (*Halder et al., 1995*). After 24 hr of feeding, only 48% of NBs were EdU-positive NBs, indicating that ectopic Ey antagonizes non-MB NB reactivation from quiescence (*Figure 4G,L*). Nonetheless, at 3 days AFW, the number of EdU-positive NBs remained relatively unchanged, whereas in controls, EdU-positive NBs declined by 31% (*Figure 4H,L,M*). Even at 7 days AFW, some non-MB NBs still incorporated EdU in *ey^{WT}*-overexpressing animals (*Figure 4J, L–N*). While ectopic Ey expression allows some non-MB NBs to proliferate longer during the initial phase of nutrient withdrawal, ectopic Ey is not sufficient to convert NB proliferation from nutrient-dependent to nutrient-independent. This could be due to technical reasons; in 24 hr fed animals overexpressing *ey^{WT}*, 70% of non-MB NBs express relatively high levels of Ey, whereas at 7 days AFW, 49% of non-MB NBs express relatively low levels of Ey (*Figure 3G,M–T*). Alternatively, there could be lineage specific-effects that account for the differences, which cannot yet be discerned.

We set out to determine whether dietary amino acids function as a cue only for NB reactivation or alternatively, whether NB subtypes have different dietary nutrient requirements for proliferation. We found that during early larval stages, most NBs exit cell cycle when dietary amino acids are withdrawn, yet a small subset, the MB NBs, continue dividing (*Britton and Edgar, 1998*; *Lin et al., 2013*). We also showed that the transcription factor Eyeless (Ey), a Pax-6 orthologue, expressed in MB NBs is required for MB NB nutrient-independent proliferation. Important future work will include the identification of Ey target genes in MB NBs. A preliminary bioinformatic analysis has revealed a number of putative Ey DNA-binding sites in regulatory regions of genes involved in metabolism (*Supplementary file 1*). In addition, the source of the amino acids that support MB NB proliferation decisions remains an open question. Amino acids must come from either extracellular sources or through catabolic recovery of amino acids within MB NBs themselves. Finally, it will be essential to determine whether other stem cell types also regulate proliferation decisions in response to nutrient availability in a lineage-dependent, cell-autonomous manner.

## Materials and methods

### Fly strains

Stocks used in this study were: *Oregon R, worGal4* (*Albertson and Doe, 2003*), *UAS-dp60* (*Weinkove et al., 1999*), *UAS-dp110^{CAAX}* (*Leevers et al., 1996*), *UAS-ey* (*Halder et al., 1995*), *UAS-*

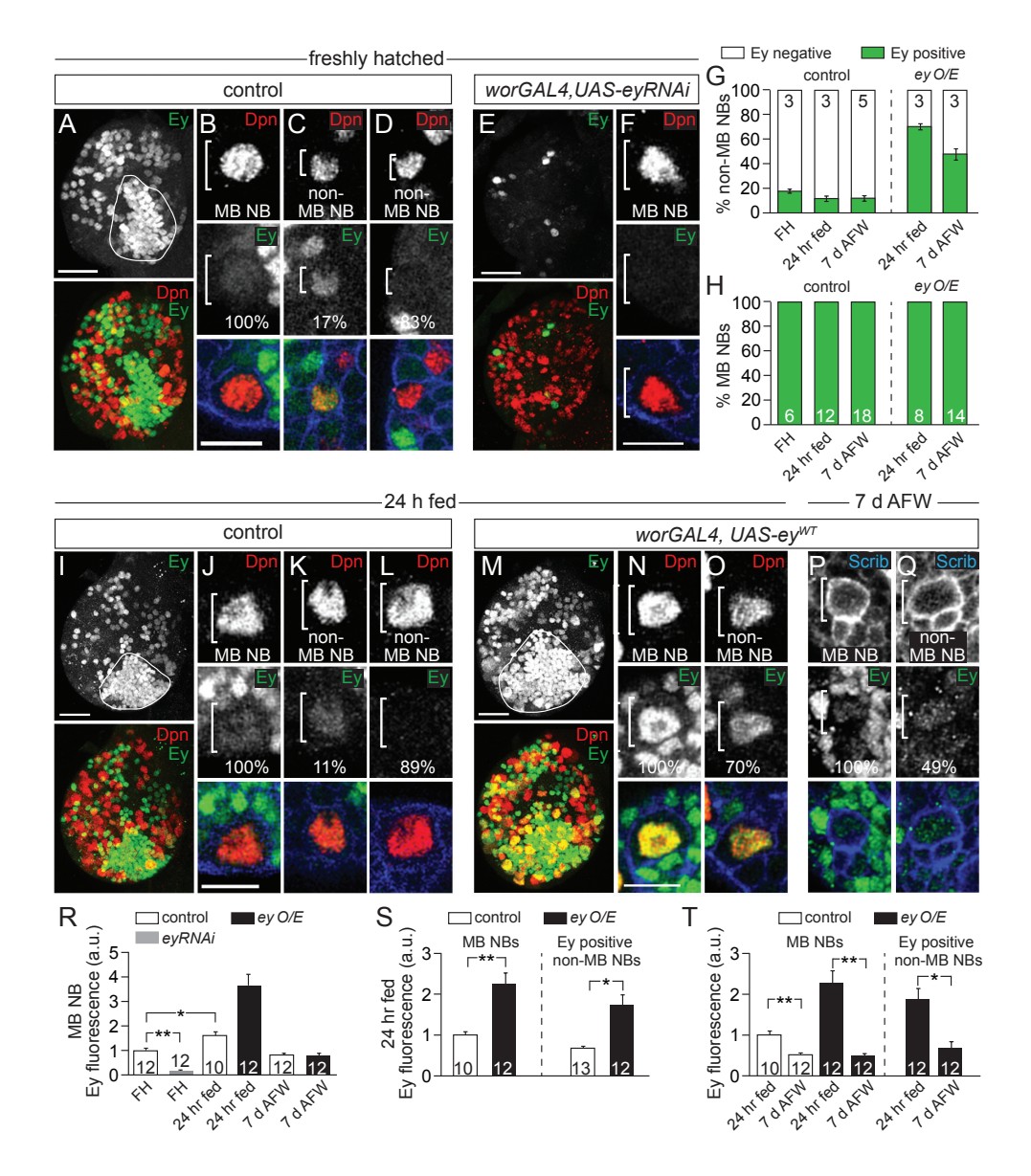

**Figure 3.** Eyeless is expressed in MB NBs. (**A,E,I,M**) Maximum intensity projections of single brain hemispheres, single-channel greyscale image in top panel with colored overlay below. Genotypes and developmental stage listed above and molecular makers listed within panels. The field of Ey-positive MB neurons outlined in white in A,I,M. (**A,E,I,M**) Scale bar equals 20 μm. (**B–D,F,J–L,N–Q**) MB NBs or non-MB NBs at higher magnification from indicated genotypes and time points. Single plane NBs marked in white brackets. (**B,F,J,N**) Scale bars equals 10 μm. (**G,H**) Percentage of non-MB NBs or MB NBs expressing Ey per brain hemisphere. Column numbers indicate number of brain hemispheres scored. (**R–T**) Quantification of average relative Ey fluorescence in MB NBs and in non-MB NBs (see Materials and methods). Column numbers equal number of NBs scored. **p-value<0.001, *p-value<0.01, two-tailed t-test, error bars, SEM.

DOI: https://doi.org/10.7554/eLife.26343.010

The following source data is available for figure 3:

**Source data 1.** Quantification of Ey expression in MB NBs and in non-MB NBs in brains of control animals and in brains of animals expressing either *UAS-eyRNAi* or *UAS-ey^{WT}*.

DOI: https://doi.org/10.7554/eLife.26343.011

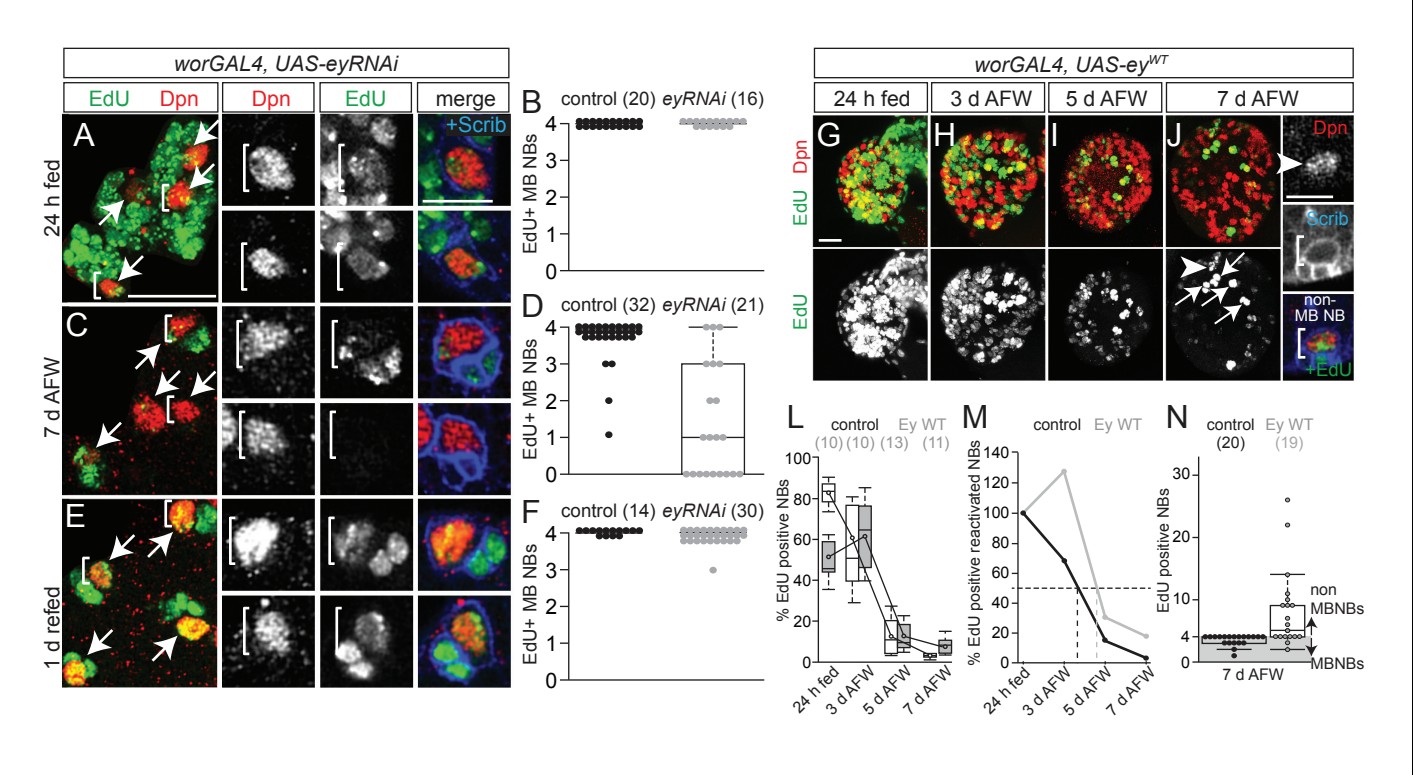

**Figure 4.** Eyeless is required for nutrient-independent MB NB proliferation. (A,C,E) Maximum intensity projections of the four MB NBs (indicated by arrows) from the indicated genotype and time points. Brackets indicate two of the four MB NBs shown at higher magnification in right panels. (B,D,F) Box plots of the number of EdU-positive MB NBs per brain hemisphere from the indicated genotypes and time points. Small circles denote primary data. Numbers in parentheses indicate the number of brain hemispheres analyzed. (G–J) Maximum intensity projections of single brain hemispheres, top panel colored overlay with single-channel greyscale image below. Genotype and time points listed above, and molecular markers to the left. Arrows in (J) indicate the four MB NBs, arrowhead marks a non-MB NB shown at higher magnification at right (white brackets). (L) Box plots of the percent EdU-positive NBs per brain hemisphere at the indicated time points. Numbers in parentheses indicate the number of brain hemispheres analyzed at each time point, color corresponds to genotype. (M) Percent of EdU-positive NBs normalized to the percent of reactivated NBs after 24 hr of feeding. Dotted lines indicate the time at which 50% of reactivated NBs are EdU-positive. (N) Box plot of the number of EdU-positive NBs in each brain hemisphere at 7 d AFW. Small circles denote primary data; those in the gray-shaded area are brain lobes in which only MB NBs are EdU-positive. Numbers in parentheses indicate number of brain hemispheres analyzed. Scale bars: A (large panel), G: 20 μm; A (small panel), K: 10 μm.

DOI: https://doi.org/10.7554/eLife.26343.012

The following source data is available for figure 4:

**Source data 1.** Quantification of EdU-positive MB NBs in brains expressing *UAS-eyRNAi* and *UAS-ey^WT* during dietary amino acid withdrawal.
DOI: https://doi.org/10.7554/eLife.26343.013

*eyRNAi* (Bloomington Stock Center, FBst0032486), *UAS-alkRNAi* (Bloomington Stock Center, FBst0027518), *pcna:GFP* (*Thacker et al., 2003*), *UAS-N-Venus-PH-GRP* (*Khuong et al., 2013*), *UAS-C-Venus-PH-GRP* (*Khuong et al., 2013*). Control animals were progeny from *worGal4* crossed to *Oregon R*. All animals were raised in uncrowded conditions at 25°C.

## Nutrient withdrawal

The complete larval diet consisted of standard Bloomington fly food. Nutrient withdrawal experimental protocol was adapted from Britton and Edgar (*Britton and Edgar, 1998*). Freshly hatched larvae were allowed to feed for 24 hr on a complete diet. Larvae were then transferred to a solution of 20% sucrose/PBS for the indicated number of days. Larvae were transferred to fresh 20% sucrose solution daily. Twenty-four hours before analysis, larvae were transferred to 20% sucrose/PBS containing 200 μM EdU. For refeeding experiments, larvae in sucrose-only for 7 days were placed back onto a complete diet for the indicated number of days and EdU was added to food for the final 24 hr. Drosophila were raised at 25°C throughout.

In *Figure 1—figure supplement 1E,F*, freshly hatched larvae were fed for 48 hr on a complete diet before switching to a sucrose-only diet.

## Immunofluorescence

Larval brains were dissected in PBS and fixed in 4% paraformaldehyde/PEM for 20 min, followed by detection of EdU using the Click-iT EdU imaging kit (Molecular Probes, Eugene, OR) (*Doyle et al., 2017*). After thorough washing in 0.1% Triton-x/PBS, antibody staining was performed according to standard methods (*Doyle et al., 2017*; *Siegrist et al., 2010*). Primary antibodies used in this study were: chicken anti-GFP (1:100, Abcam, Cambridge, MA), rat anti-Deadpan (1:100, Abcam), rabbit anti-Scribble (1:200; gift of C. Doe), and rabbit anti-Eyeless (1:1000; gift of U. Waldorf). Secondary antibodies were conjugated to Alexa Fluor dyes (Molecular Probes).

## Imaging and quantification

Z-stacks encompassing entire individual brain hemispheres were collected using a Leica SP8 laser scanning confocal microscope equipped with a $63\times$/1.4 NA oil-immersion objective. MB NBs were conclusively identified based on their stereotypical position on the dorsal surface of the brain, Ey staining, and the axonal projections of their progeny (visualized by Scrib), which extend into the calyx of the developing mushroom body. Numbers of Dpn-positive, EdU-positive, Ey-positive, and PH:Venus-positive NBs in individual brain hemispheres were counted manually using ImageJ. For cell size measurements, approximate neuroblast diameter was determined by measuring two perpendicular lines across the center of each cell in ImageJ and averaging the two lengths.

Quantification of Ey fluorescence in both NB subtypes was performed using ImageJ. Scrib immunostaining was used to outline the NB manually, generating a region of interest. Total Ey fluorescence per NB was calculated as the product of NB area and mean pixel intensity of the background corrected Ey channel. Background fluorescence measurements were acquired from nearby regions devoid of Ey expressing cells and subtracted from mean Ey fluorescence intensity. In *Figure 3*, we report normalized average fluorescence intensities across genotypes and developmental stages. These were calculated by dividing the average Ey fluorescence intensity in MB NBs by the corresponding base value averages of MB NBs and non-MB NBs at indicated time points and genotypes.

For box plots, the boundary of the box closest to zero indicates the 25th percentile, a line within the box marks the median, and the boundary of the box farthest from zero indicates the 75th percentile. Whiskers (error bars) above and below the box indicate the 90th and 10th percentiles, respectively. Data are presented in the text as ±standard error of the mean, and experimental data sets were tested for significance using two-tailed Student's t-tests in the R software package. Source data files for each experiment specify the number of animals and brain hemispheres quantified for each genotype and time point.

## Acknowledgements

We thank C. Doe, U. Waldorf, the TRiP at Harvard Medical School, and the Bloomington Stock Center for fly stocks and/or antibody reagents. We thank Karsten Siller at the University of Virginia, Advanced Research Computing Service for development of the ImageJ plugin used for fluorescence intensity measurements. We thank Karsten Siller, Chris Doe, Susie Doyle, and Matt Pahl, for comments on the manuscript, and all members of the Siegrist lab for helpful comments and discussions. This work was funded by the National Institutes of Health (NIH)/Eunice Kennedy Shriver National Institute of Child Health and Human Development (R00-HD067293), by NIH/National Institute of General Medical Sciences (R01-GM120421), and by NIH/National Institute of Neurological Disorders and Stroke (F32-NS096919).

## Additional information

### Funding

| Funder | Grant reference number | Author |
| --- | --- | --- |
| National Institute of Neurological Disorders and Stroke | F32-NS096919 | Conor W Sipe |

| Eunice Kennedy Shriver National Institute of Child Health and Human Development | R00-HD067293 | Sarah E Siegrist |
| National Institute of General Medical Sciences | R01-GM120421 | Sarah E Siegrist |

The funders had no role in study design, data collection and interpretation, or the decision to submit the work for publication.

## Author contributions
Conor W Sipe, Conceptualization, Data curation, Formal analysis, Validation, Investigation, Visualization, Methodology, Writing—original draft, Project administration, Writing—review and editing; Sarah E Siegrist, Conceptualization, Data curation, Formal analysis, Supervision, Funding acquisition, Validation, Investigation, Visualization, Methodology, Writing—original draft, Project administration, Writing—review and editing

## Author ORCIDs
Conor W Sipe (iD) http://orcid.org/0000-0002-4455-6612
Sarah E Siegrist (iD) http://orcid.org/0000-0003-0685-5387

## Decision letter and Author response
Decision letter https://doi.org/10.7554/eLife.26343.017
Author response https://doi.org/10.7554/eLife.26343.018

# Additional files

## Supplementary files
• Supplementary file 1. Candidate *Drosophila* metabolic genes regulated by Ey. To assemble the list of candidate genes, regions 1 kb upstream and 200 bp downstream of annotated *Drosophila* promoters were searched using a position weight matrix of the Ey-binding site (*Punzo et al., 2002*). This set was then compared against a pre-compiled background dataset comprised of distribution scores for promoter-binding sites across the genome, allowing a determination of the likelihood of a given site as a putative Ey regulatory element. Candidates were further refined to genes with a known function in metabolism using Gene Ontology terms.
DOI: https://doi.org/10.7554/eLife.26343.014
• Transparent reporting form
DOI: https://doi.org/10.7554/eLife.26343.015

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
