## [Decision Letter]

Thank you for submitting your article "Eyeless uncouples neuroblast proliferation from dietary amino acids in *Drosophila*" for consideration by *eLife*. Your article has been reviewed by Three peer reviewers, one of whom, Bruce Edgar is a member of our Board of Reviewing Editors and the evaluation has been overseen by the Reviewing Editor and Didier Stainier as the Senior Editor.

The reviewers have discussed the reviews with one another and the Reviewing Editor has drafted this decision to help you prepare a revised submission.

Summary:

In this report Sipe and Siegrist revisit earlier observations that *Drosophila*'s mushroom body neuroblasts (MB NB) cycle independently of larval feeding, extending this observation to show that this behavior is determined by the eyeless (ey) transcription factor and is Pi3K-independent, whereas other NB are both nutrition and Pi3K dependent for their proliferation. The demonstration that loss of eyeless in MB NBs renders them nutrition-dependent is convincing and significant. They also show that a small subset of non MB NBs can be transformed into a nutrition-insensitive state by ectopic ey, suggesting that the difference in cell cycle behavior in these neuroblasts is ey dependent. No insight is give as to why this subset of NBs are ey-dependent, and the critical target genes of ey for these cell cycle effects are not determined.

Essential revisions:

The three reviewers agreed that the demonstration of ey functions in MB NBs to make them nutrition independent is an interesting insight worthy of publication. However, the reviewers agreed that the conclusion that ey mediates this function in other NB is over-sold, as only a few NB are affected. Moreover, the reviewers agreed that the paper could be improved considerably by additional data relating to how eyeless mediates these functions, and that without such data the results don't fully warrant publication as a full-format paper. We therefore invite you to revise the manuscript as a short report (~1500 words main text, 3-4 figures) that addresses the points listed below, which are summarized from the reviews. For a full format paper, we would expect a somewhat more in-depth analysis that delved into the targets of eyeless in NBs, and how they control cell identities and these different cell cycle behaviors.

1) The main claim of the paper is that Ey is both necessary and sufficient to maintain neuroblast proliferation under nutrient restriction. The authors claim that "ectopic ey is sufficient to switch non-MB NB proliferation from nutrient-dependent to nutrient-independent" and "ey is sufficient to maintain non-MB NB proliferation in the absence of dietary amino acids" (subsection “Ectopic Ey is sufficient to switch non-MB NB proliferation from nutrient-dependent to nutrient-independent”). However, after ectopic expression of Ey in non-MB neuroblasts only a very low proportion (~4% per brain lobe) become amino acid independent. Thus, the authors cannot conclude that "Ey and PI3K are sufficient to drive non-MB NB proliferation in the absence of dietary amino acids". Furthermore, some non-MB neuroblasts express Ey endogenously. Could these neuroblasts be the ~4% that respond? Since it seems that Ey has much less effect on non MB NBs as compared to MB NBs, it would be good for the authors to try to determine what subset of non-MB NBs are affected by the ectopic Ey. In summary, the conclusion that ey is sufficient to maintain nutrient-independent cycling is not accurate, and needs to be re-stated in a more qualified form.

2) The authors over-interpret the results of some of their experiments. They only do 24hr feeding/starvation which means that they have analysed solely L1 larvae (even though 7 or 8 days old they are still at L1 stages) and from these state that non MB neuroblasts are nutrient dependent. This contradicts Britton and Edgar, 1998 and Cheng et al., 2011 who showed that after full initial neuroblast reactivation (the initial analysis were performed at 48hrs ALH=end L2; and at 60hrs ALH=early L3, respectively) non MB neuroblasts become nutrient independent. So, unless the authors test the effect of starvation in neuroblast cycling at these later time points and collect data that contradicts the earlier reports they cannot generally claim that "non-MB NBs require dietary amino acids for proliferation."

3) Figure 3 addresses Eyeless expression, but it's confusing and doesn't support the conclusions in the text. The signals in WT are very weak, and not quantified, so the conclusion that Ey is expressed in MB NB but not non-MB NBs is not well supported. In addition, Figure 3 doesn't show whether Ey is expressed in Non-MBs after nutrient activation. This information would be helpful. Overall, better data on Ey expression should be provided.

---

## [Author Response]

*Essential revisions:*

*1) The main claim of the paper is that Ey is both necessary and sufficient to maintain neuroblast proliferation under nutrient restriction. The authors claim that "ectopic ey is sufficient to switch non-MB NB proliferation from nutrient-dependent to nutrient-independent" and "ey is sufficient to maintain non-MB NB proliferation in the absence of dietary amino acids" (subsection “Ectopic Ey is sufficient to switch non-MB NB proliferation from nutrient-dependent to nutrient-independent”). However, after ectopic expression of Ey in non-MB neuroblasts only a very low proportion (~4% per brain lobe) become amino acid independent. Thus, the authors cannot conclude that "Ey and PI3K are sufficient to drive non-MB NB proliferation in the absence of dietary amino acids". Furthermore, some non-MB neuroblasts express Ey endogenously. Could these neuroblasts be the ~4% that respond? Since it seems that Ey has much less effect on non MB NBs as compared to MB NBs, it would be good for the authors to try to determine what subset of non-MB NBs are affected by the ectopic Ey. In summary, the conclusion that ey is sufficient to maintain nutrient-independent cycling is not accurate, and needs to be re-stated in a more qualified form.*

We agree with reviewers and have toned down the language and provided a more quantitative analysis of Eyeless expression in both MB NBs and in non-MB NBs over time (see Figure 3).

However, two points we would like to make:

1) Ectopic Ey expression results in 4.2 +- 1.69 EdU-positive non-MB NBs on average at 7 d AFW (revised Figure 4N). Compared to control animals, EdU-positive non-MB NBs are never observed at 7 days AFW. Although 4.2 +- 1.69 EdU-positive non-MB NBs is a relatively small number, it does equal close to 10% of reactivated non-MB NBs (43.2 on average) in Ey^wt^-overexpressing animals (Figure 4L). This relatively modest effect could be due to the inability to maintain Ey expression in non-MB NBs over the course of nutrient withdrawal. This is supported by quantitative Ey expression data now included in revised Figure 3. Alternatively, other intrinsic factors, as yet unknown, could also be required.

2) Perhaps more striking is the observation that the percent of EdU-positive NBs in Ey^wt^-overexpressing animals at 3 d AFW remains relatively constant compared to 24 hour fed animals, which in controls shows a marked decrease. The sustained proliferation of non-MB NBs in Ey^wt^-overexpressing animals during early nutrient withdrawal correlates with increases in numbers of non-MB NBs expressing Ey and higher levels of Ey in non-MB NBs.

In light of reviewer comments and additional data we provide, the text has been modified as following:

"While ectopic Ey may keep some non-MB NBs proliferating longer during early nutrient withdrawal, ectopic Ey is not sufficient to convert NB proliferation from nutrient-dependent to nutrient-independent. This could be due to technical reasons; in 24 hour fed *ey^W^*^T^-overexpressing animals, 70% of non-MB NBs express relatively high levels of Ey, whereas at 7 days AFW, only 48% of non-MB NBs express relatively low levels of Ey. Alternatively, there could be lineage specific-effects that account for the difference, which at the moment, can not be discerned."

it would be good for the authors to try to determine what subset of non-MB NBs are affected by the ectopic Ey.

This is a great idea, and one we have been thinking about as well. Unfortunately, we do not have the genetic tools to identify a specific population of non-MB NBs that may be affected by ectopic Ey expression. EdU-incorporating neuroblasts are spread throughout the brain lobe, and we are unable to detect a pattern based on their positioning. We hope to follow up on this interesting and important question in future investigations.

*2) The authors over-interpret the results of some of their experiments. They only do 24hr feeding/starvation which means that they have analysed solely L1 larvae (even though 7 or 8 days old they are still at L1 stages) and from these state that non MB neuroblasts are nutrient dependent. This contradicts Britton and Edgar, 1998 and Cheng et al., 2011 who showed that after full initial neuroblast reactivation (the initial analysis were performed at 48hrs ALH=end L2; and at 60hrs ALH=early L3, respectively) non MB neuroblasts become nutrient independent. So, unless the authors test the effect of starvation in neuroblast cycling at these later time points and collect data that contradicts the earlier reports they cannot generally claim that "non-MB NBs require dietary amino acids for proliferation."*

Thank you for this important point. We have now clarified the language to include "during early larval stages". As an example, "We conclude that during early larval stages, most NBs enter and exit cell cycle in a nutrient-dependent manner and, like developmental quiescence, this process is reversible."

In addition, we have carried out amino acid withdrawal experiments in animals after 48 hours of feeding, which are presented in Figure 1—figure supplement 1.

Finally, at no point do we compare our results with that of Cheng et al., (2011). In their experiments, larvae were nutrient restricted after animals reached critical weight, a developmental checkpoint regulated by ecdysone which commits animals to metamorphosis regardless of further nutrient intake. Thus, the physiology of NBs after the critical weight transition is likely quite different.

*3) Figure 3 addresses Eyeless expression, but it's confusing and doesn't support the conclusions in the text. The signals in WT are very weak, and not quantified, so the conclusion that Ey is expressed in MB NB but not non-MB NBs is not well supported. In addition, Figure 3 doesn't show whether Ey is expressed in Non-MBs after nutrient activation. This information would be helpful. Overall, better data on Ey expression should be provided.*

Eyeless expression in MB NBs and in non-MB NBs is now quantified in Figure 3.